# Water-powered self-propelled magnetic nanobot for rapid and highly efficient capture of circulating tumor cells

Ravindra D. Wavhale[1], Kshama D. Dhobale[1], Chinmay S. Rahane[1], Govind P. Chate[1], Bhausaheb V. Tawade[1], Yuvraj N. Patil[1], Sandesh S. Gawade[2] & Shashwat S. Banerjee[1✉]

Nanosized robots with self-propelling and navigating capabilities have become an exciting field of research, attributable to their autonomous motion and specific biomolecular interaction ability for bio-analysis and diagnosis. Here, we report magnesium (Mg)-$Fe_3O_4$-based Magneto-Fluorescent Nanorobot ("MFN") that can self-propel in blood without any other additives and can selectively and rapidly isolate cancer cells. The nanobots *viz;* Mg-$Fe_3O_4$-GSH-G4-Cy5-Tf and Mg-$Fe_3O_4$-GSH-G4-Cy5-Ab have been designed and synthesized by simple surface modifications and conjugation chemistry to assemble multiple components viz; (i) EpCAM antibody/transferrin, (ii) cyanine 5 NHS (Cy5) dye, (iii) fourth generation (G4) dendrimers for multiple conjugation and (iv) glutathione (GSH) by chemical conjugation onto one side of Mg nanoparticle. The nanobots propelled efficiently not only in simulated biological media, but also in blood samples. With continuous motion upon exposure to water and the presence of $Fe_3O_4$ shell on Mg nanoparticle for magnetic guidance, the nanobot offers major improvements in sensitivity, efficiency and speed by greatly enhancing capture of cancer cells. The nanobots showed excellent cancer cell capture efficiency of almost 100% both in serum and whole blood, especially with MCF7 breast cancer cells.

[1] Central Research Laboratory, Maharashtra Institute of Medical Education and Research, Talegaon Dabhade, Pune 410507, India. [2] Department of Surgery, Maharashtra Institute of Medical Education and Research, Talegaon Dabhade, Pune 410507, India. ✉email: shashwatbanerjee@mitmimer.com

Self-propelled micro- and nanoscale devices offer promise for diverse practical applications in biomedicine and have thus stimulated considerable research efforts in recent years[1–6]. However, most systems reported so far are inappropriate for biomedical use as they rely on fuels considered incompatible with living organisms, such as $H_2O_2$, acidic, alkaline, $Br_2$, or $I_2$ solutions for chemical propulsion[2,7]. Hence, the most important practical application of self-propelled micro- and nanobots in diverse biological systems is seriously hindered[2,7]. Therefore, nano- and micro-scale propulsion systems utilizing biocompatible and environment friendly fuel sources to power are strongly desired to eliminate the need for adding external fuels[7,8]. For practical nanobot applications, water is the obvious ideal choice of fuel compared to extreme acidic or alkaline media[5]. However, fabrication of versatile water driven micro/nanobots possessing advanced mobility and ability to perform complex biological functions, such as featuring specific cell recognitions in shortest time frame, represents an exciting yet challenging task in the field of nanobiotechnology[9].

In light of these advantages, we present smart self-propelled Mg-based Janus nanobots, which can self-propel in blood without any other additives and can selectively isolate cancer cells. The nanobots were prepared by fabricating a hemispherical shell of $Fe_3O_4$ on Mg nanoparticles and then by selectively assembling multiple components viz; (i) anti-Epithelial cell adhesion molecule (EpCAM) monoclonal antibody (Ab)/transferrin (Tf) for targeting cancer cells, (ii) cyanine 5 NHS (Cy5) dye for particle labeling, (iii) fourth generation (G4) dendrimer for multiple conjugation and (iv) glutathione (GSH) linker by chemical conjugation onto one side of Mg nanoparticle using the parafilm method. The self-propulsion of the Mg-$Fe_3O_4$-GSH-G4-Cy5-Ab/Tf (MFN) nanobot is based on the hydrogen bubbles produced by spontaneous Mg-water reaction.

Mg being a biocompatible 'green' nutrient trace element, vital for many bodily functions and enzymatic processes, makes it an attractive material for designing water-driven nanobots[10–12]. In addition, Mg is a low cost metal and is stable in ambient atmosphere as compared to other hydrogen-generating active metals like potassium, calcium, and sodium[13]. Furthermore, the incorporation of $Fe_3O_4$ shell on Mg nanoparticle enables the water-driven nanobot to be magnetically guided and functionalized to perform various important tasks.

The nanobot due to its unique features of autonomous motion and specific biomolecular interaction ability could find a potential application for detection and rapid isolation of circulating tumor cells (CTCs), in unprocessed body fluids. Due to the extremely low number of CTCs present in the blood (as few as 1 CTC per $1 \times 10^9$ hematological cells), detection and isolation is a challenge[10]. Most of current CTC isolation approaches are limited by their slow rate and low CTC-capture yield[14–16]. The continuous movement of the nanobot through the sample may offer major improvements in the sensitivity and speed of biological studies by greatly enhancing the target binding efficiency. Furthermore, the efficient cargo-towing ability of such self-propelled nanobot, along with precise motion control can lead to medical diagnostic microchips driven via sustainable endogenous chemical activity.

## Results and discussion

**Preparation and characterization of the Mg-based Janus nanobot.** The fabrication process of the Mg based Janus nanobot is schematically illustrated in Fig. 1a. In the first step, a layer of Mg nanoparticles with a diameter of ~12 nm was dispersed onto a parafilm-coated glass slide. Upon mild heating at 40 °C, the Mg nanoparticles were partially embedded into the parafilm and the remaining area of the nanoparticles was exposed for subsequent fabrication processes. This resulted in an asymmetric structure, namely the Janus robot. Specifically, the parafilm containing the embedded Mg particles were treated with a solution containing $Fe^{2+}$ and $Fe^{3+}$ salts to form the $Fe_3O_4$ shell by co-precipitation method. The hemispherical $Fe_3O_4$ shell formed is responsible for imparting magnetic property to the Mg nanosphere. Next, PAMAM G4 dendrimer was conjugated to the hemispherical $Fe_3O_4$ shell through the GSH linker using thiol chemistry to facilitate simultaneous attachment of multiple functional groups. To impart fluorescence imaging capability, Cy5 was covalently bonded with amine groups on PAMAM G4. Thereafter, anti-EpCAM antibody (Ab) was conjugated to the activated COOH group of GSH by EDC coupling reaction, resulting in Mg-$Fe_3O_4$-GSH-G4-Cy5-Ab nanobot. Similarly, Tf was conjugated to form Mg-$Fe_3O_4$-GSH-G4-Cy5-Tf, as it has been used as a model targeting moiety to the cancer cells with overexpressed Tf receptors[6,14]. Figure 1b shows the methodology utilized for cancer cell capture using the Janus nanobots. Samples containing cancer cells were mixed with the Janus nanobot and separated by application of a magnetic field.

The Janus nature of as-prepared MFN was confirmed by TEM and EDX mapping. TEM analysis of MFN showed the particles size to be ~20 nm (Fig. 2a and Supplementary Fig. S1). A binary heterostructure is obviously discernible from Fig. 2a, in which more than half of the Mg nanosphere surface is covered by an asymmetric spherical-cap of $Fe_3O_4$ layer thus indicating the dual catalytic and magnetic layer concentrated on either side of its surface. This facilitates the asymmetrical ejection of $H_2$ bubbles from the exposed surface of the Mg nanosphere in water, creating a directional propulsion thrust. Scanning transmission electron microscopy (STEM) images and STEM-Energy Dispersive X-ray Spectroscopy (EDX) mapping analysis was carried out to confirm the nanobot composition. Supplementary Fig. S2 of the resulting EDX images illustrates the presence and distribution of Mg, Fe, and O, respectively. The hydrodynamic size of MFN was analyzed to be $62 \pm 3.3$ nm (Supplementary Fig. S3).

Supplementary Fig. S4 shows MFN propelling in PBS buffer at pH 7.4 with 0.5 M $NaHCO_3$ and its response when held next to a permanent magnet. Interestingly, the MFN moving in vertical trajectory changed their direction and moved in horizontal direction under the influence of an external magnetic field as shown in the real-time tracking trajectory Fig. S4b. The MFN accumulated at the side of the tube where the magnetic field gradient was the strongest consequently indicating that the MFN direction can be remotely controlled by a magnetic field. Thus, the MFN is able to actuate in dual mode (catalytically powered by Mg nanoparticles or magnetically actuated by $Fe_3O_4$ shell), with the possibility of simultaneous guiding and steering by magnetic fields.

The MFN was also characterized by FTIR to verify the successful covalent conjugation between PAMAM G4, Cy5, and Tf/Ab. Figure 2b shows the IR spectra of Mg, Mg-$Fe_3O_4$, Mg-$Fe_3O_4$-GSH-G4, Mg-$Fe_3O_4$-GSH-G4-Cy5, Mg-$Fe_3O_4$-GSH-G4-Cy5-Tf, and Mg-$Fe_3O_4$-GSH-G4-Cy5-Ab, respectively. The IR spectrum of Mg-$Fe_3O_4$ showed prominent peaks at 578 cm$^{-1}$, 667 cm$^{-1}$ due to Fe–O stretching and a wide peak in the range 3300–3500 cm$^{-1}$, due to O−H stretching vibration thus confirming the presence of $Fe_3O_4$ shell. G4 conjugated Mg-$Fe_3O_4$ showed characteristic peaks 1640, 3384, and 3532 cm$^{-1}$ due to the amide C=O groups and N-H stretching of primary amine -$NH_2$ indicating conjugation of GSH-G4 with Mg-$Fe_3O_4$. The spectrum of Tf conjugated Mg-$Fe_3O_4$-GSH-G4-Cy5 showed new peaks at 1597 cm$^{-1}$ and 1498 cm$^{-1}$ due to CO groups from amide I and CN stretching vibration from amide II band, respectively. The presence of different functional groups such as

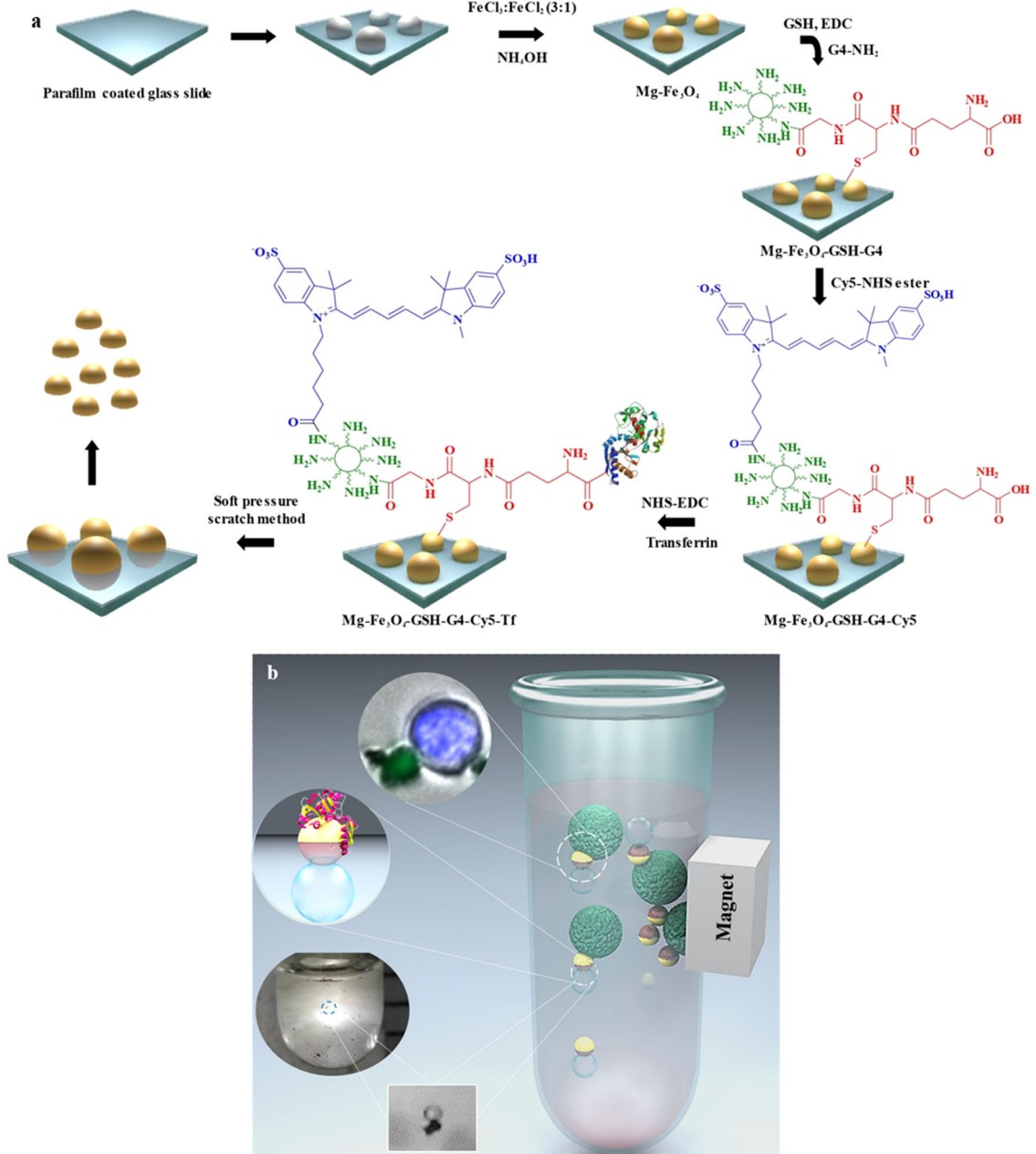

**Fig. 1 Schematic of the MFN fabrication and cancer cell capture strategy. a** Schematic illustration of the fabrication of Janus MFNs. **b** Schematic of the self-propelling MFN and HCT116 cell capture. Lower left side inset shows the upward moving nanobot due to the adhered $O_2$ bubble. Upper left side inset shows HCT116 cell captured by MFN and separated under magnetic field.

NH (1398 cm$^{-1}$) and CH (960 cm$^{-1}$) in the spectrum also confirms the presence of Tf conjugation. In case of Ab conjugated Mg-Fe$_3$O$_4$-GSH-G4-Cy5, the new peaks observed at 1632, 1331, 1107, and 858 cm$^{-1}$ confirmed conjugation of Ab to Mg-Fe$_3$O$_4$-GSH-G4-Cy5[17,18].

We also evaluated the conjugation reaction with respect to the change in zeta potential of the individual step during the synthesis of MFN (Fig. 2e). The zeta potentials of Mg-Fe$_3$O$_4$, Mg-Fe$_3$O$_4$-GSH-G4, Mg-Fe$_3$O$_4$-GSH-G4-Cy5, and Mg-Fe$_3$O$_4$-GSH-G4-Cy5-Ab were determined to be −4.39, −2.55, −0.442, and −6.51 mV, respectively. The step-wise altered zeta potentials indicated successful conjugation of the multiple components with Mg-Fe$_3$O$_4$. The presence of Cy5 in Mg-Fe$_3$O$_4$-GSH-G4 was further confirmed by fluorescence spectroscopy methods. The fluorescence spectrum of MFN was compared with the spectra of

Cy5 under identical optical conditions ($\lambda_{ex} = 600$ nm) (Supplementary Fig. S5). As depicted in Supplementary Fig. S5, the spectra of MFN displayed the typical emission band for Cy5 indicating conjugation of Cy5-NHS.

**Propulsion performance of MFN nanobot**. The autonomous motility of nanobot was tracked in varying media including PBS, Dulbecco's modified eagle medium (DMEM), and serum to validate biological compatibility and average velocity was measured across each medium (Fig. 3). The propulsion data were acquired by optical tracking of individual nanobot aggregate samples. Figure 3 and S6 illustrates real-time tracking trajectories of MFN in biologically relevant media with NaHCO$_3$. The nanobots propelled upward instantaneously by generation of strong thrust and buoyancy due to the H$_2$ bubble and gradually

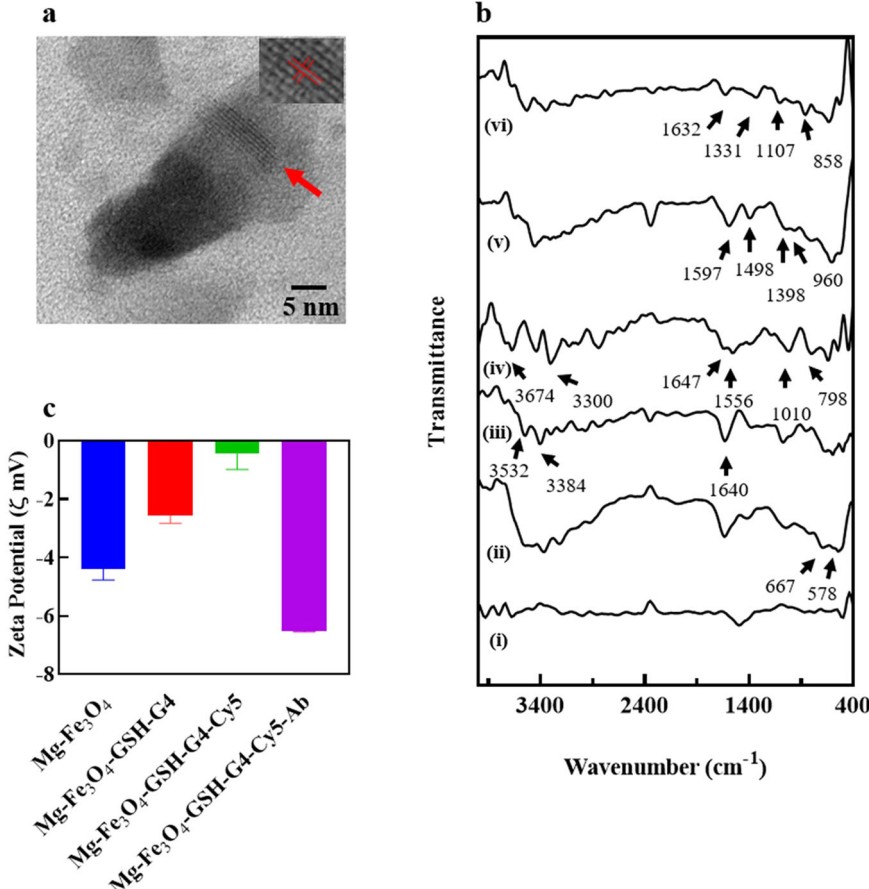

**Fig. 2 Characterization study of MFN. a** TEM image of self-propelling Janus nanobot wherein the Mg nanoparticle forms the core covered by superparamagnetic shell of $Fe_3O_4$. The crystalline features and the identified lattice fringes co-related well to the structure of magnetite planes with a plane-to-plane separation of 0.249 nm (inset). **b** FTIR spectra of (i) Mg, (ii) Mg-$Fe_3O_4$, (iii) Mg-$Fe_3O_4$-GSH-G4 and (iv) Mg-$Fe_3O_4$-GSH-G4-Cy5, (v) Mg-$Fe_3O_4$-GSH-G4-Cy5-Tf, and (vi) Mg-$Fe_3O_4$-GSH-G4-Cy5-Ab. **c** Surface charge evolution upon conjugation of GSH-G4, Cy5, and Ab.

reverted in the downward direction (Fig. 3 and Supplementary Movie 1) due to gravity once the bubble dispersed. In the meantime, more $H_2$ bubbles formed and adhered to the MFN nanobot. The $H_2$ bubble adhered to the nanobot grew larger by coalescence of several smaller bubbles. When the overall volume of the bubble was sufficiently high, the buoyancy force balanced the gravitational and viscous forces and the nanobot again moved upward.

The highly efficient locomotion of the MFN nanobot involved $H_2$ bubbles generated from the spontaneous redox reaction of Mg and water as illustrated in Fig. 3 and the following Eq. (1):[19]

$$Mg_{(s)} + 2H_2O_{(aq)} = Mg(OH_2)_{2(s)} + H_{2(g)} + \Delta_r^\theta G \qquad (1)$$

Hydrogen bubbles of around 10 µm diameter generated on one side of the nanobot is clearly observed, reflecting the rapid spontaneous reaction of Mg with the surrounding water (Fig. 1b). The nascent bubbles produce a strong momentum that propelled the nanobot upward with speeds of $0.815 \pm 0.086$ mm s$^{-1}$, $0.707 \pm 0.06$ mm s$^{-1}$, and $0.393 \pm 0.07$ mm s$^{-1}$ in PBS, DMEM, and serum, respectively, in the presence of 1.0 M $NaHCO_3$. This represents a large driving force of over 0.13 pN in PBS, based on the drag force $F = 6pmrv$, where r is the radius of the nanobot, v is the speed, and m is the viscosity of the aqueous medium.

Figure 3d shows the time-lapse images of the MFN Janus nanobot in water, indicating a cyclic motion in upward and downward direction. The average life span of the nanobot is about 8 min, 9.1 min, and >28 min in PBS, DMEM, and serum at 1.0 M $NaHCO_3$. The small opening due to the hemispherical

$Fe_3O_4$ shell enables a controlled reaction process and gradual dissolution of the Mg core, leading to a prolonged nanobot lifetime. Prudently, the presence of $NaHCO_3$ plays a vital role in self-propulsion of the MFN. The locomotion of the MFN due to hydrogen bubbles produced from the reaction of Mg and water also leads to the formation of a passivation layer of $Mg(OH)_2$ on the exposed Mg surface. Rapidly removing the $Mg(OH)_2$ passivation layer on the exposed Mg surface is crucial for accelerating Mg-water reaction to produce hydrogen propulsion for the Janus nanobot. Presence of $NaHCO_3$ can reliably assist bubble propulsion as it rapidly dissolves $Mg(OH)_2$ precipitates. The related chemical reactions are given in Eq. (2):[10]

$$Mg(OH)_2 + 2HCO_3^- \rightleftharpoons MgCO_3 + CO_3^{2-} + 2H_2O \qquad (2)$$

Further, the catalytic ability of the MFN evaluated in PBS, DMEM, and blood serum comprising a range of $NaHCO_3$ (0 to 1.0 M) concentrations revealed increased rate of reaction with increase in $NaHCO_3$ concentration. The speed of MFN increased from $0.457 \pm 0.073$ mm s$^{-1}$ to $0.815 \pm 0.086$ mm s$^{-1}$, $0.468 \pm 0.07$ mm s$^{-1}$ to $0.707 \pm 0.06$ mm s$^{-1}$, and $0.329 \pm 0.10$ mm s$^{-1}$ to $0.393 \pm 0.07$ mm s$^{-1}$ when the $NaHCO_3$ concentration is increased from 0.25 M to 1.0 M.

Interestingly, the MFN demonstrated propulsion in DMEM and serum even in the absence of externally added $NaHCO_3$. This is because DMEM and serum contain low levels of the $HCO_3$ anion. However, the increase in speed of MFN with $NaHCO_3$ concentration was found to be less for DMEM and serum as compared to PBS because of the presence of proteins in DMEM and serum. Protein fouling hinders the Mg-water reaction and

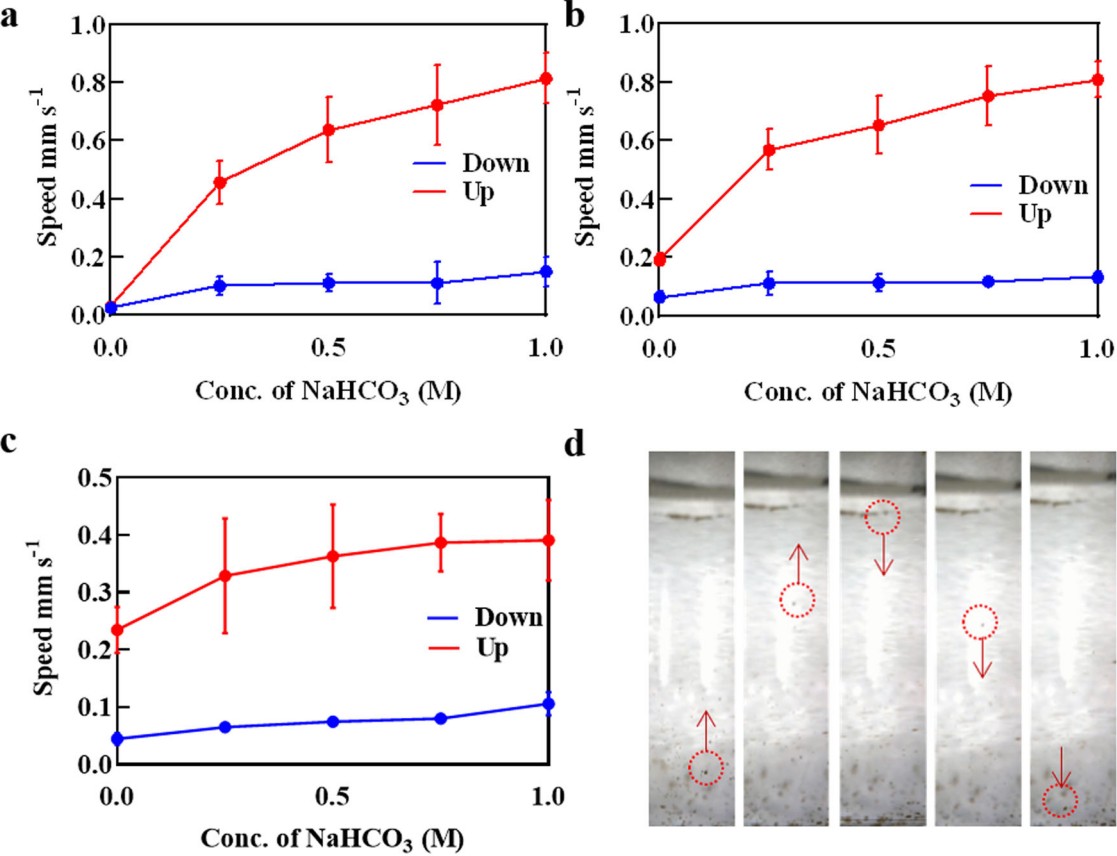

**Fig. 3 Propulsion kinetics of MFN.** Analysis of upward and downward propulsion speed of the nanobot in: **a** PBS, **b** DMEM, **c** serum, **d** time-lapse images of nanobot in 0.5 M NaHCO$_3$ solution in PBS.

subsequently the generation of hydrogen bubbles and propulsion efficiency. However, MFN speed observed in serum is sufficient for conducting various tasks in biological environments.

In addition to MFN kinetics, total propulsion time of MFN in the presence and absence of NaHCO$_3$ was also determined (Supplementary Table S1). Without the addition of NaHCO$_3$, while PBS predictably showed lack of MFN propulsion activity, DMEM and serum showed propulsion. This is exaggerated in serum with activity time almost seven times that of DMEM, attributable to endogenous NaHCO$_3$ as a result of serum carbonic anhydrase activity. In the presence of 1.0 M NaHCO$_3$, MFN propulsion time is pronounced, with serum showing almost 4-fold longer propulsion time. While this may be a result of endogenous NaHCO$_3$ in blood samples, the presence of serum carbonic anhydrase may prolong MFN propulsion.

**Isolation performance of Mg-based Janus nanobots in artificial samples.** After having evaluated the motion properties of MFN, we tested their application in effectively capturing and isolating cancer cells. To better understand the capture efficiency of Ab-modified Janus nanobot and Tf-modified Janus nanobot for HCT116 (colon cancer) cells, the study was performed under two different conditions, viz: PBS and DMEM. Cell suspensions with varying number of HCT116 cells were incubated with MFN for 5 min. Figure 4 shows a group of representative fluorescent images for captured HCT116 cells from whole blood, using MFN nanobot. Figure 4a shows an HCT116 cell captured with a Cy5 conjugated nanobot (Fig. 4b) and Fig. 4c shows a composite image of differential staining of HCT116 and leukocytes capture from whole blood.

Both the MFNs viz. Mg-Fe$_3$O$_4$-GSH-G4-Cy5-Tf and Mg-Fe$_3$O$_4$-GSH-G4-Cy5-Ab demonstrated capture efficiency greater than 93% in PBS and DMEM (Fig. 5a, b). Furthermore, the MFNs showed capture efficiency of ~100% at a very low CTC count of 10 cells mL$^{-1}$. Thus, the MFNs demonstrated excellent capture efficiency for HCT116 cells not only in PBS but also in DMEM. However, in the absence of NaHCO$_3$, the capture efficiency of MFN was less. It was only ~38, 40, 44, and 35% in PBS, DMEM, whole blood, and lysed blood for CTC count of 10 cells mL$^{-1}$ (Supplementary Table S2). Furthermore, we confirmed the HCT116 cell viability in DMEM containing 1.0 M NaHCO$_3$ at 10, 30, and 60 min (Supplementary Fig. S7). The study showed the cells were viable with no apparent cell death even after 60 min (Supplementary Figs. S7a–d).

To explore the capture sensitivity of the MFNs for scarce CTC, artificial CTC blood samples were prepared by spiking Hoechst dye stained HCT116 cells into healthy human blood with cell densities of approximately 10, 20, 50, and 100 cells mL$^{-1}$, respectively. The spiked blood samples were treated with MFNs with or without NaHCO$_3$. In whole blood and lysed blood in the presence of NaHCO$_3$, the MFNs displayed capture efficiency greater than ~98% (Fig. 5a, b). In samples with low numbers of CTCs, the capture efficiency was ~100%. Further, the capture efficiency of the MFNs was found to be higher in blood as compared to PBS and DMEM. This could be due to the inherent presence of NaHCO$_3$ in human blood[20], which propelled the MFNs and allowed it to capture the HTC116 cells more efficiently. The capture efficiency of the MFNs in whole blood and lysed blood in the absence of NaHCO$_3$ was around ~67% or less in samples with low numbers of CTCs. Thus, the capture efficiency of MFN in whole blood and lysed blood with NaHCO$_3$

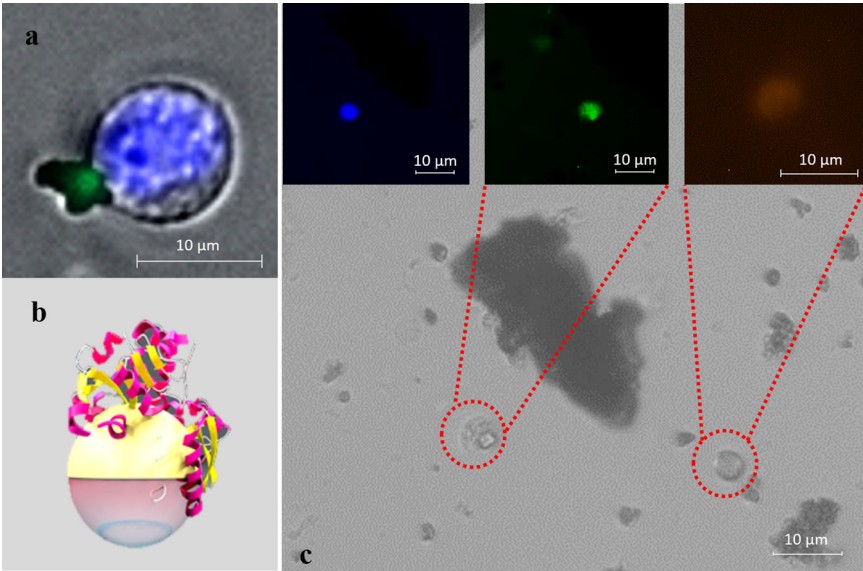

**Fig. 4 Images of cancer cells captured by MFN from blood samples. a** HCT116 cell captured by MFN conjugated with Cy5. Cell visualized by nuclear staining with Hoechst reagent (blue) and Cy5 (green). **b** Schematic of the MFN conjugated with Tf/Ab. **c** Immunocytochemistry method based on Fluorescein Isothiocyanate-labeled anti-cytokeratin (green), and Hoechst (blue) nuclear staining was applied to identify and enumerate CTCs captured by MFNs. PE-labeled anti-CD45 (red) and Hoechst (blue) nuclear staining was used to identify leukocytes.

was significantly higher as compared to whole blood and lysed blood without NaHCO3. The study confirms that the continuous motion of MFNs through the sample plays a significant role in cancer cell capture as they greatly enhance interaction and binding efficiency. Furthermore, in order to confirm that the specific interaction of MFNs with artificial CTCs (HCT116 cells) is attributable to Ab and Tf, we used nanobot without Ab and Tf to capture HCT116 cells. The nanobot demonstrated capture efficiency of ~17% after 5 min of incubation, suggesting a significant role of Ab and Tf.

To test the recognition performance of the MFNs against other cell types, MCF7 (breast cancer) cells were chosen for epithelial CTC capture. The capture efficiency of MFNs for MCF7 cells were ~100%, in PBS, DMEM, and blood, respectively (Fig. 5). The MFNs exhibited excellent specific capture capacity toward MCF7 targeted cells, thus demonstrating efficient capture of epithelial cancer cells. Also, for MCF7, the capture efficiency of the MFNs in the absence of NaHCO3 was around ~69% or less in samples with low numbers of CTCs. Furthermore, the nanobots demonstrated ~100% cancer cell capture within 5 min of incubation whereas for other nanosystems the efficiency of cancer cell capture was lower whereas the incubation time required for efficient capture of cancer cells was much higher (Supplementary Table S3).

A comparative analysis of the MFNs in the presence and absence of NaHCO3 for both HCT116 and MCF7 cells across all treatments reveals two primary findings. The most important finding here is that NaHCO3 is shown to distinctly influence the cell capture process. The trendlines as shown in Fig. 5c depict in all treatments and in both cell groups, the NaHCO3-assisted MFNs have a significantly higher slope (red tracing) than the MFNs (blue tracing) in the absence of NaHCO3. Interestingly, the capture efficiency (defined here as no. of cells captured/no. of cells spiked) is driven to unity (1.000) in most cases where NaHCO3 is utilized. In stark comparison, without NaHCO3, the capture efficiency is only 0.5 (50%).

Secondly, the linearity of the data points corresponding to the cell capture is represented in all treatments and cell groups. More specifically, the linearity of the NaHCO3 treatment groups

(averaging ~0.999) implicates the reliability of the capture mechanism driven by target binding ligands as well as the propulsion/enhanced molecular collision machinery. In the absence of NaHCO3, as the capture efficiency drops significantly lower, the linearity is affected as well suggesting an impact on the target-binding kinetics.

As shown in Supplementary Table S2 (table shows capture efficiency—ratio of cell captured: cells spiked, all media, all cell types), the calculated capture efficiencies (CE) are consistently maintained at 1, suggesting complete and total capture of all spiked cells. In clinical context, this would translate to highly efficient rounding up of the elusive CTCs. The working range of 10–100 cells was used for statistical purposes, rather than a reflection of potential CTC load in average cancer patients, which is about 1–2 cells mL$^{-1}$ of whole blood. A mechanistic inference that is apparent from the results is that while the CE is highly dependent on cell load and binding kinetics for unassisted NPs, NaHCO3 assistance overcomes these limitations by allowing maximum interaction of NPs with hematological cells present in the sample. The mechanism thus implicates autonomous propulsion as a vital feature of the nanobots, without which the cancer cell capture and detection efficiency would be 50% or lower. Additionally, this report serves as a cautionary tale for studies relying exclusively on targeting ligands for capturing and isolating targets in biological systems.

**Clinical analysis**. Clinical validation of the CTC capture was conducted on peripheral blood samples from epithelial cancer patients under the above optimized cell-capture conditions (Supplementary Table S4). In each assay, 1.0 mL blood was incubated with MFNs for 5 min and then isolated by magnetic separation. Subsequently, a commonly used three-color immunocyto-chemistry method to identify CTCs from non-specifically captured white blood cells (WBCs), using FITC-labeled anti-Cytokeratin (CK, green), PE-labeled anti-CD45 (red), and Hoechst (blue) for staining. As shown in Fig. 6, CTCs are Hoechst +/CK+/CD45− cells. CTCs were captured with counts ranging from 1 to 5 CTCs per 1 mL blood sample of cancer patients.

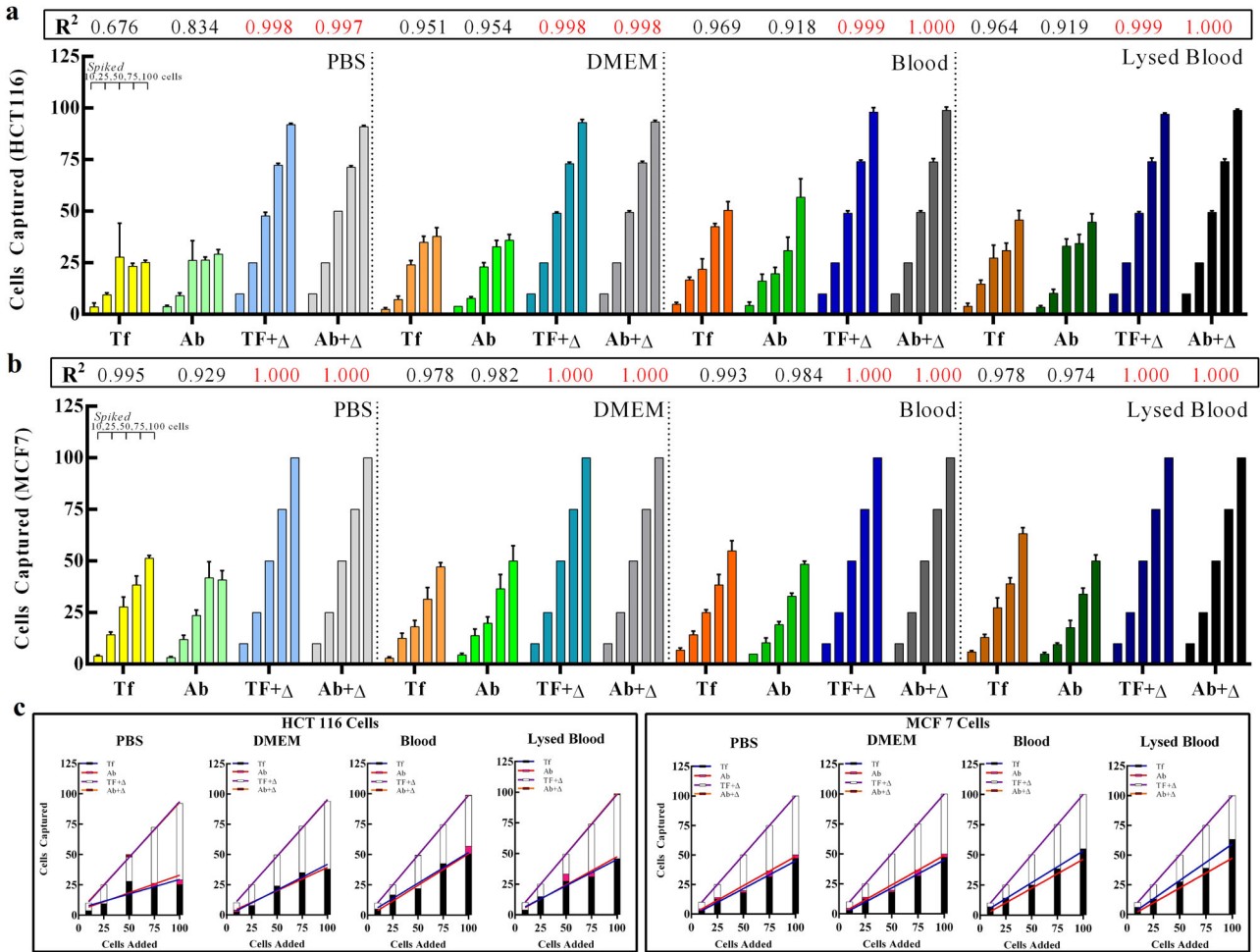

**Fig. 5 Capture efficiencies of MFNs in the presence of NaHCO₃ compared to controls. a** Grouped plots of HCT 116 cell capture by the two MFNs under control conditions and in the presence of NaHCO₃, denoted by Δ. The MFNs are treated with four media, PBS, DMEM, Blood and Lysed Blood (simulated serum) as reported here. The MFNs are further exposed to incremental quantities of spiked HCT 116 cells in an identical manner across all treatments, as indicated by the addition bar (10, 25, 50, 75, 100 cells) depicted in the top left of the plot area. **b** Depicts a similar experimental outcome using MCF7 cells under identical conditions. Data is reported as mean ± SD cells with correlation of group data represented as $R^2$ above individual plots. **c** Linear regression treatment of the data shown in **a** and **b**; HCT116 cell capture trials displayed in the left panel depicts treatments with and without the added NaHCO₃ resulting in two grouped correlation with a very high degree of linearity ('goodness of fit', $R^2 > 0.676$). The panel for MCF7 cells (right) also depict a similar outcome with more pronounced linearity ($R^2 > 0.929$). The $p$ values for statistical comparison were calculated as being less than 0.05 across the entire treatment panel for both cell groups.

Captured cells demonstrated intact cell nuclei and excellent CK18 staining, with no PE fluorescence. This clinical validation proved that our method can be easily translated for actual samples and offers a low cost rapid diagnostic method for cancer prognosis.

## Conclusion

In this work, we have presented water-powered Mg-based Janus nanobots capable of autonomous propulsion in PBS, DMEM or blood without the need for external fuels or fields. The nanobots were prepared by growing a hemispherical shell of Fe₃O₄ on Mg nanoparticles and then by selectively assembling multiple components such as anti-EpCAM antibody/transferrin (Tf), cyanine 5 NHS (Cy5) dye, fourth generation (G4) dendrimers and (v) glutathione (GSH) by chemical conjugation onto one side of Mg nanoparticle. The small opening due to the hemispherical Fe₃O₄ shell enables a controlled reaction process and gradual dissolution of the Mg core, leading to a prolonged motor lifetime. The potential application of the nanobots has been demonstrated toward rapid capture and isolation of CTCs in biologically

relevant media. The nanobots exhibited ~100% capture efficiency in blood with low numbers of CTCs indicating enhanced target binding efficiency due to continuous motion of the nanobots. Furthermore, Ab or Tf can be easily replaced with other antibodies by the surface modification strategy, suggesting that the proposed nanobots also hold a promising potential for in-vitro separation and detection of CTCs which are not epithelial based, toxicants or proteins.

## Methods

**Chemicals and reagents.** Magnesium nanoparticles with APS < 12 nm and purity of >99% were procured from Intelligent Materials Pvt. Ltd. (India). Ferric chloride tetrahydrate, ferrous chloride hexahydrate, transferrin (Tf), N-(3-Dimethylami-nopropyl)-N′-ethylcarbodiimide (EDC.HCl), glutathione (GSH), PAMAM–NH₂-G4 (G4) dendrimer 10 wt% in methanol (Mw 14214.7 Da, 64 end groups), N-hydroxysuccidimide (NHS), N,N-Diisopropylethylamine (DIEA), sodium bicarbonate (NaHCO₃), phosphate buffer saline (PBS), Hoechst 33250 dye, were purchased from Sigma-Aldrich, USA. McCoy's 5 A medium, fetal bovine serum (FBS), and penicillin and streptomycin (Pen-Strep) were purchased from Sigma-Aldrich, USA. Anti-epithelial cell adhesion molecule (EpCAM) antibody was purchased from Bioss Antibodies, USA. Dulbecco's Modified Eagle's medium (DMEM) and trypsin were purchased from Himedia, India. Cyanine 5 mono NHS

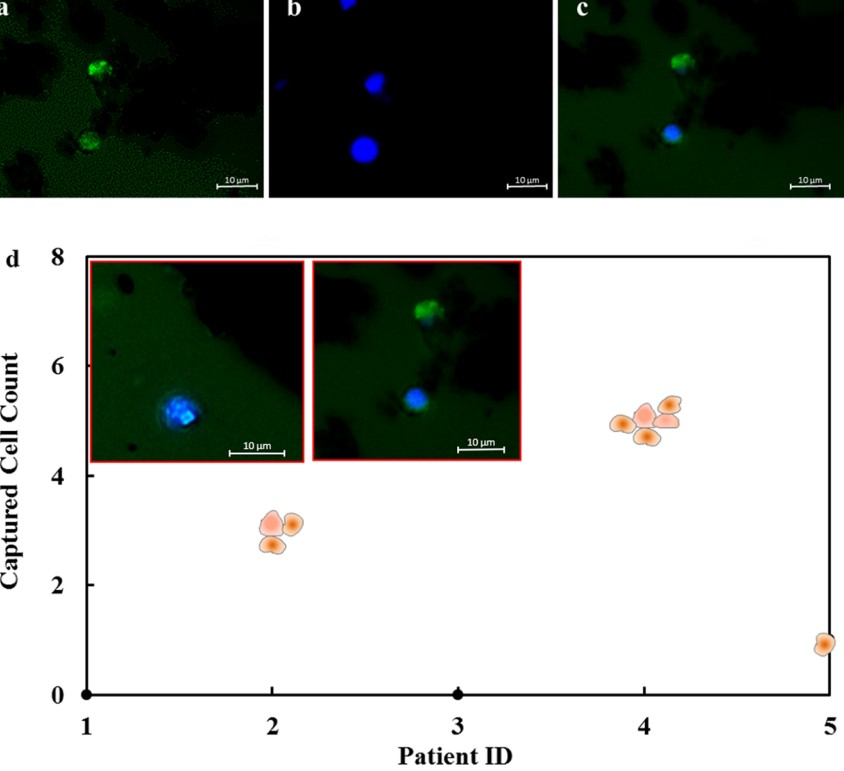

**Fig. 6 Fluorescent micrographs of CTCs captured from blood samples from a breast cancer patient.** Three-color immunocytochemistry method based on **a** FITC-labeled anti-Cytokeratin, and **b** Hoechst nuclear staining. **c** Composite image from FITC and Hoechst micrographs. **d** Plot of CTCs captured by MFN in 1 mL peripheral blood samples from epithelial cancer patients.

ester was procured from GE Healthcare, UK. Ultrapure water (MilliQ) used in this work was acquired from Direct-Q® 3 water purification system, Merck Millipore, Germany. All other chemicals were of analytical grade and used without further purification.

**Synthesis of Fe₃O₄-coated Mg nanoparticles (Mg-Fe₃O₄).** One side of the glass slide was uniformly covered with parafilm. 20 mg of Mg nanoparticles were partially embedded in the parafilm-coated glass slide by placing it in an oven at 50 °C for 2 min. The coated glass slide was tapped gently to remove loosely held Mg nanoparticles. $FeCl_3·6H_2O$ and $FeCl_2·4H_2O$ solution (2:1) was then applied to the exposed side of the Mg nanoparticles and pH of the applied solution was then adjusted to ~10 using $NH_4OH$. The glass slide with the nanoparticles was then washed 3 times with MilliQ water and dried overnight under vacuum.

**Conjugation of GSH-G4 with Mg-Fe₃O₄.** 8.0 mg of PAMAM–$NH_2$-G4 (G4) was added to 1.0 mg of glutathione (GSH) dissolved in 0.5 mL of MilliQ water having pH 6.0 in the presence of EDC.HCl and DMAP as catalysts. The reaction was continuously stirred for 2 h at room temperature on a magnetic stirrer to obtain GSH-G4 conjugate. The obtained GSH-G4 conjugate (~540 µL) was applied 3 times on Mg-Fe₃O₄ nanoparticles present on the glass slide and allowed to stand for 2 h. The nanoparticles were washed with MilliQ water for 3 times to remove excess GSH-G4. The glass slide of Mg-Fe₃O₄ nanoparticles conjugated with GSH-G4 was then dried overnight under vacuum.

**Conjugation of Cy5 with Mg-Fe₃O₄-GSH-G4.** 50 µL of 100 ppm Cy5 NHS solution in MilliQ water was mixed with 100 µL of 1000 ppm DIPEA and the final solution pH was adjusted to 7.8. The surface of Mg-Fe₃O₄-GSH-G4 was then exposed to the reaction mixture and allowed to stand for 2 h in dark. The Mg-Fe₃O₄-GSH-G4 nanoparticles on the glass slide were washed with MilliQ water for 3 times to remove excess Cy5, then dried at room temperature under vacuum, and stored in dark.

**Conjugation of Tf to Mg-Fe₃O₄-GSH-G4-Cy5.** The glass slide with Mg-Fe₃O₄-GSH-G4-Cy5 nanoparticles was treated with 0.5 mL solution of EDC.HCl (2.5 mg ml⁻¹) and 0.5 mL solution of N-Hydroxysuccinimide (NHS) (2.5 mg ml⁻¹), in phosphate buffered saline (PBS) for 1 h at room temperature. The resulting N-hydroxysuccinamide activated system was washed for 3 times with MilliQ water to remove excess EDC, NHS. Further, the activated Mg-Fe₃O₄-GSH-G4-Cy5

nanoparticles were treated with Tf (2 mg mL⁻¹) and allowed to stand for 4 h. The glass slide was washed 3 times and then dried under vacuum. Conjugation of anti-EpCAM Ab (10 µg) to Mg-Fe₃O₄-GSH-G4-Cy5 nanoparticles was carried out in a similar manner and conjugated nanoparticles were stored in HEPES buffer solution (pH 8.5).

**Cell culture.** HCT116 and MCF7 cells were procured from the National Center for Cell Sciences (NCCS) repository. HCT116 cells were cultured in McCoy's 5 A medium, supplemented with 10% fetal bovine serum and 100 unit mL⁻¹ penicillin, 100 µg mL⁻¹ streptomycin, and maintained in $CO_2$ incubator at 37 °C and 5% $CO_2$ saturation. MCF7 cells were cultured in Eagle's Minimum Essential Medium (EMEM) supplemented with 10% fetal bovine serum and 100 unit mL⁻¹ penicillin, 100 µg mL⁻¹ streptomycin, and maintained in $CO_2$ incubator at 37 °C and 5% $CO_2$ saturation.

**Characterization.** TEM analysis was performed using Tecnai FEI G2 (accelerating voltage of 300 kV). The samples were prepared by placing a drop of MFN suspension (in MilliQ water) onto a Formvar-covered copper grid. The suspension was allowed to dry in air at room temperature before imaging. Hydrodynamic diameter was determined by dynamic light scattering (DLS) using HORIBA Scientific Nano Particle Analyzer SZ-100. FTIR spectra were recorded using a Perkin Elmer Fourier Transform Infrared (FTIR) spectrometer, USA in the range between 4000 and 400 cm⁻¹, with a resolution of 2 cm⁻¹. The UV–Vis absorption studies were carried out on Agilent Technologies Cary 60 UV spectrophotometer. Fluorescence analysis was carried out using an Agilent Cary Eclipse fluorescence spectrophotometer in the range between 500 and 750 cm⁻¹ at an excitation wavelength of 650 nm. The zeta potential values were measured using Zetasizer Nano ZS (Malvern Instruments, Worcestershire, UK). Zeta (ξ) potential analysis was conducted using phase analysis light scattering mode at room temperature.

**Nanobot tracking.** The autonomous motion of the MFN nanobot (500 µg mL⁻¹) in PBS, DMEM, and serum containing varying concentration of $NaHCO_3$ (0.1, 0.25, 0.5, 0.75, and 1 M), was recorded with Dino-lite digital microscope at 50X magnification, using the DinoCapture 2.0 software. MFN used in the study were mostly in the form of aggregates as dispersion treatments such as sonication were avoided to prevent denaturation of the Tf protein and also for accurate observation and recording with a microscope of 50X magnification.

**Estimation of capture efficiency from artificial CTC suspension**. Capture efficiency of MFNs were estimated for both HCT116 and MCF7 cells under 4 different conditions: in 1X PBS, DMEM, whole blood with RBCs lysed, and whole blood without RBC lysis. Whole blood from healthy individual volunteers was collected into sterile heparinized vacutainer tubes. For RBC lysis, blood and RBC lysis buffer (1:3) were mixed gently with a sterile pipette in a 1.5 mL centrifuge tube. The blood–RBC lysis buffer mixture was incubated for 10 min at room temperature and centrifuged at 3000 rpm for 5 min. Supernatant was removed and cell pellet was resuspended in 100 μL PBS.

Artificial CTC samples were prepared by spiking PBS or DMEM or whole blood (with and without RBC lysis) with Hoechst-labeled HCT116 or MCF7 cells. 500 μg MFN was then added to 200 μL of the CTC sample with and without 0.5 M NaHCO$_3$. The suspension was incubated for 2 min and then subjected to strong magnetic field separation for 10 min. The captured cell pellet and the residual cell suspension were imaged and the number of HCT116 or MCF7 cells were counted.

**Validation of CTC capture in cancer patients**. In order to clinically validate the cell capture efficiency of the nanobots, CTC-capture studies were performed on peripheral blood samples from five cancer patients with metastatic tumors, including colorectal, lung, breast, and ovarian cancer patients under the above optimized cell-capture conditions. Samples were obtained after informed consent through MIMER Institutional Ethics Committee-approved protocol. Cancer patient samples were processed in the same manner as the artificial CTC samples. Post magnetic separation, the captured cell pellet was washed twice with PBS and stained with CD45-PE and CK18-FITC antibodies (Bioss, USA) for 2 h at room temperature. Post incubation, the samples were washed with PBS, labeled with Hoechst and imaged under fluorescence microscope.

## Data availability

The data generated during this study are included in the published article and the Supplementary Information.

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

## Acknowledgements

The authors acknowledge the financial support from Department of Science and Technology (SR/NM/NS-1189/2016), Government of India and Department of Biotechnology (BT/PR21922/NNT/28/1241/2017), Government of India.

## Author contributions

S.S.B. conceived and designed the project. R.D.W., G.P.C., and B.V.T. prepared the nanobots and also performed the motion experiments. K.D.D. performed the cancer cell capture experiments in artificial samples. C.S.R. performed the clinical analysis. S.S.G. supported the clinical studies. S.S.B., R.D.W., K.D.D., C.S.R., G.P.C., B.V.T., and Y.N.P. co-analyzed the experimental and calculated data. The manuscript was written by S.S.B., C.S.R., and Y.N.P. S.S.B. supervised the project. All authors reviewed the manuscript.

## Competing interests

The authors declare no competing interests.
