## [Peer Review File · Communications Chemistry]

Reviewers' comments:

Reviewer #1 (Remarks to the Author):

In this manuscript, the authors presented water-powered Mg-based Janus nanobots capable of autonomous propulsion in PBS, DMEM or blood without the need for external fuels or fields, and their use for detection of CTCs. The results presented are generally interesting and scientifically sound. However, some results are lacking and I do have some major concerns regarding the translation of this method into clinical settings.

1. Nanobots are generally studied or used for in vivo diagnosis and/or treatment of cancer in patients. However, in this manuscript, the only clinical experiment the authors have provided is the ex vivo testing of peripheral blood samples for CTCs. Generally, nanobots are over-engineered for ex vivo clinical tests as many simple nanoparticles can perform as well for ex vivo tests. If the authors intend to use the nanobots in vivo, no in vivo study is found in the manuscript to verify the effectiveness and safety of the Mg-based Janus nanobots.

2. What are the major advantages of using this Water-Powered Self-Propelled Magnetic Nanobot (as described in this manuscript) compared to currently available methods in detecting CTCs (such as the methods listed in review articles PMID: 28393954 and 33850563)? If this is just a fancy method with a much higher cost for the patients but with no major advantage, it will unlikely be translated into a clinical method. Please discuss this within the manuscript.

Reviewer #2 (Remarks to the Author):

The authors reported a kind of magnesium (Mg)-Fe₃O₄-based Magneto-Fluorescent Nanorobot (MFN) that can self-propel in blood without any other additives and can selectively and rapidly isolate cancer cells. However, part of the data in the paper did not achieve the expected effect, and some of the data were inconsistent with the descriptions in the article. Based on the above factors, I think it is not suitable for publication in Communications Chemistry. The specific comments are as follows:

1. The authors mentioned in the abstract that "the nanobot offers major improvements in sensitivity, efficiency and speed by greatly enhancing capture of cancer cells", comparing the sensitivity, efficiency and speed of the nanomotor reported in this paper with those of the magnesium-based nanomotor reported in other literature will make the results more convincing.
2. The size of nanoparticles in the Fig. 2a named as "self-propelling Janus nanobot wherein the Mg nanoparticle forms the core covered by superparamagnetic shell of Fe₃O₄" does not match the description in this paper that the Mg nanoparticles with a diameter of ~12 μm. The author should demonstrate each step of the modification shown in Fig. 1 by using characterization of SEM, TEM, or fluorescence, and so on. What is the approximate size of Fe₃O₄? It is not convincing that the TEM provides only a single particle, images of multiple particles should be provided.
3. According to the capture mechanism provided by the authors, the MFN will be gradually consumed during movement, leaving behind a shell of Fe₃O₄. Will cancer cells ingest the Fe₃O₄ shells into the cell? In addition, can the small size of Fe₃O₄ achieve the separation of cancer cells at the micron scale? It is more likely that small sized Fe₃O₄ shells will drop from cancer cells during separation process using magnets, the authors should design experiments to rule out this possibility.
4. The schematic diagram in Fig. 1b is not accurate. Cause the magnesium nanosphere will be

consumed during the movement, the incomplete MFN should be binding to the tumor cell in the end.

5. The original image that can reveal the morphology and size of the MFN in Supplementary Figure S1 should be provided, instead of just providing elemental mapping. In addition, the scale bar in Figure S1b-1e is 100 nm, which means that the size of the whole figure is about 200 nm, which cannot reflect the complete morphology and element distribution of the MFN. The complete morphology should be characterized by SEM-mapping.

6. The data of Dynamic light scattering (DLS) of nanobots should be provided.

7. The movement of the Nanobots in different media (especially in blood) should be provided in the form of video (not less than 20 nanobot in each media). Clearly marked trajectories of the each nanobot should be provided. And the detailed motion analysis of the nanobots such as MSD should be provided as well.

8. The pictures in Figure 4 and 6 are of poor quality, and only one or two cancer cells are provided. And the background interference is too difficult to judge whether cancer cells have been successfully captured accurately. Pictures with higher quality and more cancer cells should be provided.

9. The language and grammar should be further polished.

Reviewer #3 (Remarks to the Author):

In the manuscript, the authors reported cancer cell isolation in biological media using nanobots. By taking advantage of the self-propelling nature of the Mg/Fe₃O₄ Janus nanoparticles in water-based media, the authors have achieved excellent cancer cell capture and isolation efficiencies, even with the clinical samples. Although the topic is interesting, the novelty of the manuscript falls short for Communications Chemistry. The method described in this study does not show significant progress, compared to previous work of others on water-powered nanoparticle propulsion and nanobot-based cancer cell capture. I would recommend publication only with the following concerns addressed:

1. It is not clear what makes this work distinct from the previous studies on nanobot-based biomolecule capture, transfer, and isolation. The authors should articulate this point with a concrete comparison with recent studies, including (1) *Small* 14, 1704252; and (2) *Angew. Chem. Int. Ed.* 50, 4161.

2. Magnetic direction control was roughly described. Although the authors stated the nanobots changed their moving direction when a magnetic field was applied, their trajectories were not presented in either Fig. 3 or S2. Moreover, it is unclear whether the magnetic field just steered the nanobots or overrode the bubble propulsion.

3. In Fig.3, why were the nanobots thrust upward, not in a random direction? In general, the nanobots' Mg side where bubbles form should be randomly oriented, which eventually resulted in random motions. I am also wondering if it is possible to propel the nanobots downward? If it is not because buoyancy is the dominant driving mechanism, the nanobots would only have a restricted range of motion.

4. On page 6, it is uncertain the driving force was generated from a single nanobot or a nanobot aggregate. Additionally, the authors should indicate in the main body that the nanobot aggregates were used in the experiments, although it is mentioned in the method.

5. The size of Mg nanoparticles " $\sim 12 \mu\text{m}$ " on page 3 is misleading.

Water-Powered Self-Propelled Magnetic Nanobot for Rapid and Highly Efficient Capture of Circulating Tumor Cells

Submission ID: COMMSCHEM-21-0190

Response to referees' comments

Reviewer #1

In this manuscript, the authors presented water-powered Mg-based Janus nanobots capable of autonomous propulsion in PBS, DMEM or blood without the need for external fuels or fields, and their use for detection of CTCs. The results presented are generally interesting and scientifically sound. However, some results are lacking and I do have some major concerns regarding the translation of this method into clinical settings.

Response: We thank the reviewer for appreciating the work presented in the manuscript. Indeed the comment from reviewer for our manuscript is encouraging, that the “The results presented are generally interesting and scientifically sound”.

1. Nanobots are generally studied or used for in vivo diagnosis and/or treatment of cancer in patients. However, in this manuscript, the only clinical experiment the authors have provided is the ex vivo testing of peripheral blood samples for CTCs. Generally, nanobots are over-engineered for ex vivo clinical tests as many simple nanoparticles can perform as well for ex vivo tests. If the authors intend to use the nanobots in vivo, no in vivo study is found in the manuscript to verify the effectiveness and safety of the Mg-based Janus nanobots.

Response: The nanobots discussed in the present work have been developed for “liquid biopsy” i.e. CTC capture and isolation from blood withdrawn from cancer patients and the procedure is only to be performed ex-vivo. Hence, the nanobots have been developed especially for ex-vivo application and we have presented clinical data to support the viability of the nanobots for this application. We have now deleted the sentence “Furthermore, the H₂ bubbles produced from the MFN have a diameter similar to that of RBCs, indicating that they may exhibit comparable rheology in microvessels and capillaries to RBCs, and thus may not have any adverse effects to human body for their potential in-vivo applications” from the manuscript as it may have created an impression that the nanobots were to be used for in vivo applications.

2. What are the major advantages of using this Water-Powered Self-Propelled Magnetic Nanobot (as described in this manuscript) compared to currently available methods in detecting CTCs (such as the methods listed in review articles PMID: 28393954 and 33850563)? If this is just a fancy method with a much higher cost for the patients but with no major advantage, it will unlikely be translated into a clinical method. Please discuss this within the manuscript.

Response: The major advantage of the self-propelled Mg-based Janus nanobot presented in the work:

i. To the best of my knowledge, water driven nanosized robots/motors have not been reported for CTC capture. The nanobots reported in the present work have been designed by integrating multiple components to impart multifunctionality to the nanobot such as: autonomous propulsion ability in complex biological fluids, magnetic property for guidance and separation, and ability to selectively isolate cancer cells.

ii. The autonomous motion in the sample imparts the nanobots with ability to capture and isolate CTCs rapidly and selectively in a very short time. In our study we have shown that the nanobots can capture ~100% cancer cells with just 5 min of incubation. In all other reported studies the incubation time required is much more (some recent reference given below). We have now added a sentence in page 10, line 231 and the sentence read as:

“Furthermore, the nanobots demonstrated ~100% cancer cell capture within 5 min of incubation whereas for other nanosystems the incubation time required for efficient capture of cancer cells was much higher.^{21-25,}”

21. Wang *et al.*, High-Efficiency Isolation and Rapid Identification of Heterogeneous Circulating Tumor Cells (CTCs) Using Dual-Antibody-Modified Fluorescent-Magnetic Nanoparticles, *ACS Appl. Mater. Interfaces* 2019, 11, 39586–39593.

22. Meng *et al.*, Biomimetic Immunomagnetic Nanoparticles with Minimal Nonspecific Biomolecule Adsorption for Enhanced Isolation of Circulating Tumor Cells, *ACS Appl. Mater. Interfaces* 2019, 11, 28732-28739.

23. Wang *et al.*, Antifouling hydrogel-coated magnetic nanoparticles for selective isolation and recovery of circulating tumor cells, *J. Mater. Chem. B*, 2021,9, 677-682.

24. Huang *et al.*, Gelatin Nanoparticle-Coated Silicon Beads for Density-Selective Capture and Release of Heterogeneous Circulating Tumor Cells with High Purity, *Theranostics* 2018, 8, 1624-1635.

25. Dong *et al.*, Enhanced Capture and Release of Circulating Tumor Cells Using Hollow Glass Microspheres with Nanostructured Surface, *Nanoscale* 2018, 10, 16795-16804.

We have earlier reported several nanosystems for rapid and efficient CTC capture and some of the platforms based on reported design are currently being used by medical fraternity for cancer diagnosis.

Banerjee *et al.*, Self-propelled carbon nanotube based microrockets for rapid capture and isolation of circulating tumor cells, *Nanoscale*, 2015 7(19), 8684-8.

Banerjee et al., Transferrin-Mediated Rapid Targeting, Isolation, and Detection of Circulating Tumor Cells by Multifunctional Magneto-Dendritic Nanosystem, *Adv. Healthcare Mater.* 2013, 2, 800-805.

Reviewer #2

1. *The authors mentioned in the abstract that " the nanobot offers major improvements in sensitivity, efficiency and speed by greatly enhancing capture of cancer cells ", comparing the sensitivity, efficiency and speed of the nanomotor reported in this paper with those of the magnesium-based nanomotor reported in other literature will make the results more convincing.*

Response: The authors have demonstrated that the autonomous motion in the sample imparts the nanobots with ability to capture and isolate CTCs rapidly and selectively in a very short time. In study we have reported that the nanobots captured ~100% cancer cells with 5 min of incubation. In all other reported studies the incubation time required is much more (some recent reference given below). The autonomous motion in the sample imparts the nanobots with ability to capture and isolate CTCs rapidly and selectively in a very short time. In our study we have shown that the nanobots can capture ~100% cancer cells with just 5 min of incubation. In all other reported studies the incubation time required is much more (some recent reference given below). We have now added a sentence in page 10, line 231 and the sentence read as:

“Furthermore, the nanobots demonstrated ~100% cancer cell capture within 5 min of incubation whereas for other nanosystems the incubation time required for efficient capture of cancer cells was much higher.²¹⁻²⁵”

21. Wang *et al.*, High-Efficiency Isolation and Rapid Identification of Heterogeneous Circulating Tumor Cells (CTCs) Using Dual-Antibody-Modified Fluorescent-Magnetic Nanoparticles, *ACS Appl. Mater. Interfaces* 2019, 11, 39586–39593.

22. Meng *et al.*, Biomimetic Immunomagnetic Nanoparticles with Minimal Nonspecific Biomolecule Adsorption for Enhanced Isolation of Circulating Tumor Cells, *ACS Appl. Mater. Interfaces* 2019, 11, 28732-28739.

23. Wang *et al.*, Antifouling hydrogel-coated magnetic nanoparticles for selective isolation and recovery of circulating tumor cells, *J. Mater. Chem. B*, 2021,9, 677-682.

24. Huang *et al.*, Gelatin Nanoparticle-Coated Silicon Beads for Density-Selective Capture and Release of Heterogeneous Circulating Tumor Cells with High Purity, *Theranostics* 2018, 8, 1624-1635.

25. Dong *et al.*, Enhanced Capture and Release of Circulating Tumor Cells Using Hollow Glass Microspheres with Nanostructured Surface, *Nanoscale* 2018, 10, 16795-16804.

2. *The size of nanoparticles in the Fig. 2a named as “self-propelling Janus nanobot wherein the Mg nanoparticle forms the core covered by superparamagnetic shell of Fe₃O₄” does not match*

the description in this paper that the Mg nanoparticles with a diameter of ~12 μm . The author should demonstrate each step of the modification shown in Fig. 1 by using characterization of SEM, TEM, or fluorescence, and so on. What is the approximate size of Fe_3O_4 ? It is not convincing that the TEM provides only a single particle, images of multiple particles should be provided.

Response: The authors thank the reviewer for pointing out the error. Actually it is '12 nm' and not '12 μm '. We have now corrected it in the manuscript.

Further, the reviewer suggested that each step of the Mg nanoparticle surface modification should be characterized by SEM, TEM, or fluorescence, and so on. We have given TEM image. (STEM) images and STEM-Energy Dispersive X-ray Spectroscopy (EDX) mapping analysis of Mg- Fe_3O_4 to confirm the presence of asymmetric spherical-cap of Fe_3O_4 on Mg nanoparticle. For further modification, we have used techniques such as Zeta potential measurement, FTIR analysis, fluorescence spectroscopy methods to confirm conjugation of the components such as: PAMAM-G4, Cy5 and EpCam antibody mAb or Transferrin (Tf) on Mg nanoparticle since they are considered better techniques for conjugation chemistry as specified in the literature given below.

Mourdikoudis *et al.*, Characterization techniques for nanoparticles: comparison and complementarity upon studying nanoparticle properties, *Nanoscale*, 2018, 10, 12871-12934.

3. According to the capture mechanism provided by the authors, the MFN will be gradually consumed during movement, leaving behind a shell of Fe_3O_4 . Will cancer cells ingest the Fe_3O_4 shells into the cell? In addition, can the small size of Fe_3O_4 achieve the separation of cancer cells at the micron scale? It is more likely that small sized Fe_3O_4 shells will drop from cancer cells during separation process using magnets, the authors should design experiments to rule out this possibility.

Response: In the cancer cell capture experiments, the incubation time of the samples with nanobots is only 5 min. The time is too short for the Mg nanoparticle to dissolve. Further, the small opening due to the hemispherical Fe_3O_4 shell enables a controlled reaction process and makes the dissolution of the Mg core more gradual. We have also discussed in the manuscript that the average propulsion life span of the nanobot in serum at 1.0M NaHCO_3 due to the presence of Mg core in the nanobot is quite high (>28 min). Additionally, to validate the cancer cell capture and isolation by nanobots, we performed clinical studies on actual cancer patient blood samples. Figure 4 and 6 shows both fluorescence and bright field images of nanobots incubated with blood samples for 5 min and then isolated by magnetic separation. The images clearly show presence of nanobot with captured cancer cells attached to it confirming the ability of the nanobots to selectively and efficiently capture rare cancer cells.

4. The schematic diagram in Fig. 1b is not accurate. Cause the magnesium nanosphere will be consumed during the movement, the incomplete MFN should be binding to the tumor cell in the end.

Response: As mentioned in the experimental section and explained in the response for query 4, the incubation time of MFNs with samples is only 5 min. Figure 4 and 6 shows the presence of nanobot with captured cancer cells attached to it confirming the ability of the nanobots to selectively and efficiently capture rare cancer cells.

5. The original image that can reveal the morphology and size of the MFN in Supplementary Figure S1 should be provided, instead of just providing elemental mapping. In addition, the scale bar in Figure S1b-1e is 100 nm, which means that the size of the whole figure is about 200 nm, which cannot reflect the complete morphology and element distribution of the MFN. The complete morphology should be characterized by SEM-mapping.

Response: As suggested by the reviewer, we have now included the Scanning transmission electron microscopy (STEM) image of MFN with corresponding elemental maps in Supplementary Figure S1.

6. The data of Dynamic light scattering (DLS) of nanobots should be provided.

Response: We have now added DLS data in the manuscript (page 4, line 92). The sentence with the DLs data now read as: ‘The hydrodynamic size of MFN was analyzed to be 62 ± 3.3 nm.’

7. The movement of the Nanobots in different media (especially in blood) should be provided in the form of video (not less than 20 nanobot in each media). Clearly marked trajectories of the each nanobot should be provided. And the detailed motion analysis of the nanobots such as MSD should be provided as well.

Response: As suggested by the reviewer, we have now included real time tracking trajectories of multiple MFNs in PBS, DMEM and Blood serum in the supporting information section (Figure S4).

Supplementary Figure S4. Analysis of the motion behavior of MFN. Representative tracking trajectories of MFNs in different biologically relevant media.

8. *The pictures in Figure 4 and 6 are of poor quality, and only one or two cancer cells are provided. And the background interference is too difficult to judge whether cancer cells have been successfully captured accurately. Pictures with higher quality and more cancer cells should be provided.*

Response: As suggested by the reviewer, we have now added images of high quality with less background interference. We have mentioned in the manuscript that only 1 to 5 CTCs per 1 mL blood sample of cancer patients were found. Hence, we could only show very few cancer cells in Figure 4 and 6.

9. *The language and grammar should be further polished.*

Response: As suggested by the reviewer we have done the necessary corrections.

Reviewer #3

In the manuscript, the authors reported cancer cell isolation in biological media using nanobots. By taking advantage of the self-propelling nature of the Mg/Fe₃O₄ Janus nanoparticles in water-based media, the authors have achieved excellent cancer cell capture and isolation efficiencies, even with the clinical samples. Although the topic is interesting, the novelty of the manuscript falls short for Communications Chemistry. The method described in this study does not show significant progress, compared to previous work of others on water-powered nanoparticle propulsion and nanobot-based cancer cell capture. I would recommend publication only with the following concerns addressed:

Response: The authors thank the reviewer for recommending publication of the manuscript with corrections.

1. *It is not clear what makes this work distinct from the previous studies on nanobot-based biomolecule capture, transfer, and isolation. The authors should articulate this point with a concrete comparison with recent studies, including (1) Small 14, 1704252; and (2) Angew. Chem. Int. Ed. 50, 4161.*

Response: In the article ‘Angew. Chem. Int. Ed. 50, 4161’ a microrocket has been reported, developed by anchoring anti-CEA mAb on Ti/Fe/Au/Pt microtube for capture of cancer cells. In the present work, for the first time to the best of my knowledge, a water driven nanosized robots/motors have been reported for CTC capture. The nanobots have been designed by integrating multiple components to impart multifunctionality to the nanobot such as: autonomous propulsion ability in complex biological fluids, magnetic property for guidance and ability to selectively isolate cancer cells. Further, we have reported that the nanobots are able to capture ~100% cancer cells in just 5 min of incubation. This is possible only due to the autonomous motion of the nanobots in the sample which enables them to capture and isolate CTCs in a very short time. In all other reported studies the incubation time required is much more (some recent reference given below). We have now added a sentence in page 10, line 231 and the sentence read as:

“Furthermore, the nanobots demonstrated ~100% cancer cell capture within 5 min of incubation whereas for other nanosystems the incubation time required for efficient capture of cancer cells was much higher.²¹⁻²⁵”

21. Wang *et al.*, High-Efficiency Isolation and Rapid Identification of Heterogeneous Circulating Tumor Cells (CTCs) Using Dual-Antibody-Modified Fluorescent-Magnetic Nanoparticles, *ACS Appl. Mater. Interfaces* 2019, 11, 39586–39593.

22. Meng *et al.*, Biomimetic Immunomagnetic Nanoparticles with Minimal Nonspecific Biomolecule Adsorption for Enhanced Isolation of Circulating Tumor Cells, *ACS Appl. Mater. Interfaces* 2019, 11, 28732-28739.

23. Wang *et al.*, Antifouling hydrogel-coated magnetic nanoparticles for selective isolation and recovery of circulating tumor cells, *J. Mater. Chem. B*, 2021,9, 677-682.

24. Huang *et al.*, Gelatin Nanoparticle-Coated Silicon Beads for Density-Selective Capture and Release of Heterogeneous Circulating Tumor Cells with High Purity, *Theranostics* 2018, 8, 1624-1635.

25. Dong *et al.*, Enhanced Capture and Release of Circulating Tumor Cells Using Hollow Glass Microspheres with Nanostructured Surface, *Nanoscale* 2018, 10, 16795-16804.

2. *Magnetic direction control was roughly described. Although the authors stated the nanobots changed their moving direction when a magnetic field was applied, their trajectories were not presented in either Fig. 3 or S2. Moreover, it is unclear whether the magnetic field just steered the nanobots or overrode the bubble propulsion.*

Response: As suggested by reviewer, we have now given the trajectory of the nanobot when magnetic field is applied in Supplementary Figure S2. b. The tracking trajectory of MFN in PBS buffer with 0.5M NaHCO₃ (left) clearly confirms the discussion we have given in the manuscript (page 4): “Supplementary Fig. S2 shows MFN propelling in PBS buffer at pH 7.4 with 0.5M NaHCO₃ and its response when held next to a permanent magnet (right). Interestingly, the MFN moving in vertical trajectory changed their direction and moved in horizontal direction under the influence of an external magnetic field. The MFN accumulated at the side of the tube where the magnetic field gradient was the strongest consequently indicating that the MFN direction can be remotely controlled by a magnetic field.”

3. *In Fig.3, why were the nanobots thrust upward, not in a random direction? In general, the nanobots' Mg side where bubbles form should be randomly oriented, which eventually resulted in random motions. I am also wondering if it is possible to propel the nanobots downward? If it is not because buoyancy is the dominant driving mechanism, the nanobots would only have a restricted range of motion.*

Response: We have discussed in the manuscript that the nanobots propelled upward instantaneously by generation of buoyancy due to the bubble adhered to it and then gradually reverted in the downward direction due to gravity once the bubble dispersed. In the meantime,

more H₂ bubbles formed and adhered to the nanobot. The H₂ bubble adhered to the nanobot grew larger by coalescence of several smaller bubbles. When the overall volume of the bubble was sufficiently high, the buoyancy force balanced the gravitational and viscous forces and the nanobot again moved upward. The motion mechanism has been discussed in detail in our earlier reported articles on nanobots:

Banerjee et al., Self-Propelling Targeted Magneto-Nanobots for Deep Tumor Penetration and pH-Responsive Intracellular Drug Delivery, *Scientific Reports* 2020, 10, 4703.

Banerjee et al., Self-propelled carbon nanotube based microrockets for rapid capture and isolation of circulating tumor cells, *Nanoscale*, 2015 7(19), 8684-8688.

4. *On page 6, it is uncertain the driving force was generated from a single nanobot or a nanobot aggregate. Additionally, the authors should indicate in the main body that the nanobot aggregates were used in the experiments, although it is mentioned in the method.*

Response: As suggested by the reviewer, we have mentioned nanobot aggregate and the sentence on page 6 (line 131) now read as: ‘The propulsion data were acquired by optical tracking of individual nanobot aggregate samples.’

5. *The size of Mg nanoparticles "~12 um" on page 3 is misleading.*

Response: We thank the reviewer for pointing out the error. We have corrected it to ‘12 nm’.

Reviewers' comments:

Reviewer #1 (Remarks to the Author):

Thank you very much for the detailed response to the reviewers. However, one of my previous concerns that "nanobots are overengineered for ex vivo clinical tests as many simple nanoparticles can perform as well for ex vivo tests" has not been addressed. I think the authors can consider doing a simple experiment to compare the effectiveness of nanoparticles with no propulsion and the nanobots in CTC capture to check if there is significant benefit in using nanobots.

Reviewer #2 (Remarks to the Author):

I insist on my last point of view. Please refer to the attachment for the item-by-item response to the author's reply.

Reviewer #2

1. The authors mentioned in the abstract that " the nanobot offers major improvements in sensitivity, efficiency and speed by greatly enhancing capture of cancer cells ", comparing the sensitivity, efficiency and speed of the nanomotor reported in this paper with those of the magnesium-based nanomotor reported in other literature will make the results more convincing.

Response: The authors have demonstrated that the autonomous motion in the sample imparts the nanobots with ability to capture and isolate CTCs rapidly and selectively in a very short time. In study we have reported that the nanobots captured ~100% cancer cells with 5 min of incubation. In all other reported studies the incubation time required is much more (some recent reference given below). The autonomous motion in the sample imparts the nanobots with ability to capture and isolate CTCs rapidly and selectively in a very short time. In our study we have shown that the nanobots can capture ~100% cancer cells with just 5 min of incubation. In all other reported studies the incubation time required is much more (some recent reference given below). We have now added a sentence in page 10, line 231 and the sentence read as:

“Furthermore, the nanobots demonstrated ~100% cancer cell capture within 5 min of incubation whereas for other nanosystems the incubation time required for efficient capture of cancer cells was much higher.²¹⁻²⁵”

21. Wang *et al.*, High-Efficiency Isolation and Rapid Identification of Heterogeneous Circulating Tumor Cells (CTCs) Using Dual-Antibody-Modified Fluorescent-Magnetic Nanoparticles, *ACS Appl. Mater. Interfaces* 2019, 11, 39586–39593.

22. Meng *et al.*, Biomimetic Immunomagnetic Nanoparticles with Minimal Nonspecific Biomolecule Adsorption for Enhanced Isolation of Circulating Tumor Cells, *ACS Appl. Mater. Interfaces* 2019, 11, 28732-28739.

23. Wang *et al.*, Antifouling hydrogel-coated magnetic nanoparticles for selective isolation and recovery of circulating tumor cells, *J. Mater. Chem. B*, 2021,9, 677-682.

24. Huang *et al.*, Gelatin Nanoparticle-Coated Silicon Beads for Density-Selective Capture and Release of Heterogeneous Circulating Tumor Cells with High Purity, *Theranostics* 2018, 8, 1624-1635.

25. Dong *et al.*, Enhanced Capture and Release of Circulating Tumor Cells Using Hollow Glass Microspheres with Nanostructured Surface, *Nanoscale* 2018, 10, 16795-16804.

Comment : It would be better to compare the nanobots' speed, capture efficiency and concentration of cell in this paper with other researches in the form of Table, and it not sufficient to compare the capture time simply. In addition, it is meaningless to compare capture efficiency without providing the concentration of cell. At the same time, the paper lacks the detailed characterization of nanobots' speed.

2. The size of nanoparticles in the Fig. 2a named as “self-propelling Janus nanobot wherein the Mg nanoparticle forms the core covered by superparamagnetic shell of Fe_3O_4 ” does not match the description in this paper that the Mg nanoparticles with a diameter of $\sim 12 \mu m$. The author should demonstrate each step of the modification shown in Fig. 1 by using characterization of SEM, TEM, or fluorescence, and so on. What is the approximate size of Fe_3O_4 ? It is not convincing that the TEM provides only a single particle, images of multiple particles should be provided.

Response: The authors thank the reviewer for pointing out the error. Actually it is ‘12 nm’ and not ‘12 μm ’. We have now corrected it in the manuscript.

Further, the reviewer suggested that each step of the Mg nanoparticle surface modification should be characterized by SEM, TEM, or fluorescence, and so on. We have given TEM image. (STEM) images and STEM-Energy Dispersive X-ray Spectroscopy (EDX) mapping analysis of Mg- Fe_3O_4 to confirm the presence of asymmetric spherical-cap of Fe_3O_4 on Mg nanoparticle. For further modification, we have used techniques such as Zeta potential measurement, FTIR analysis, fluorescence spectroscopy methods to confirm conjugation of the components such as: PAMAM-G4, Cy5 and EpCam antibody mAb or Transferrin (Tf) on Mg nanoparticle since they are considered better techniques for conjugation chemistry as specified in the literature given below.

Mourdikoudis *et al.*, Characterization techniques for nanoparticles: comparison and complementarity upon studying nanoparticle properties, *Nanoscale*, 2018, 10, 12871-12934.

Comment: Although the data of Zeta and FTIR was used to demonstrate each step of the modification, it still lack of the TEM figure contains multiple particles.

3. According to the capture mechanism provided by the authors, the MFN will be gradually consumed during movement, leaving behind a shell of Fe_3O_4 . Will cancer cells ingest the Fe_3O_4 shells into the cell? In addition, can the small size of Fe_3O_4 achieve the separation of cancer cells at the micron scale? It is more likely that small sized Fe_3O_4 shells will drop from cancer cells during separation process using magnets, the authors should design experiments to rule out this possibility.

Response: In the cancer cell capture experiments, the incubation time of the samples with nanobots is only 5 min. The time is too short for the Mg nanoparticle to dissolve. Further, the small opening due to the hemispherical Fe_3O_4 shell enables a controlled reaction process and makes the dissolution of the Mg core more gradual. We have also discussed in the manuscript that the average propulsion life span of the nanobot in serum at 1.0M $NaHCO_3$ due to the presence of Mg core in the nanobot is quite high (>28 min). Additionally, to validate the cancer cell capture and isolation by nanobots, we performed clinical studies on actual cancer patient blood samples. Figure 4 and 6 shows both fluorescence and bright field images of nanobots

incubate with blood samples for 5 min and then isolated by magnetic separation. The images clearly show presence of nanobot with captured cancer cells attached to it confirming the ability of the nanobots to selectively and efficiently capture rare cancer cells.

4. *The schematic diagram in Fig. 1b is not accurate. Cause the magnesium nanosphere will be consumed during the movement, the incomplete MFN should be binding to the tumor cell in the end.*

Response: As mentioned in the experimental section and explained in the response for query 4, the incubation time of MFNs with samples is only 5 min. Figure 4 and 6 shows the presence of nanobot with captured cancer cells attached to it confirming the ability of the nanobots to selectively and efficiently capture rare cancer cells.

Comment: How long is the life of the nanobots in this paper? will the nanobots deplete within 5 minutes? It would be more reasonable to provide the video which record the corresponding motion of the nanobots.

5. *The original image that can reveal the morphology and size of the MFN in Supplementary Figure S1 should be provided, instead of just providing elemental mapping. In addition, the scale bar in Figure S1b-1e is 100 nm, which means that the size of the whole figure is about 200 nm, which cannot reflect the complete morphology and element distribution of the MFN. The complete morphology should be characterized by SEM-mapping.*

Response: As suggested by the reviewer, we have now included the Scanning transmission electron microscopy (STEM) image of MFN with corresponding elemental maps in Supplementary Figure S1.

6. *The data of Dynamic light scattering (DLS) of nanobots should be provided.*

Response: We have now added DLS data in the manuscript (page 4, line 92). The sentence with the DLs data now read as: 'The hydrodynamic size of MFN was analyzed to be 62 ± 3.3 nm.'

Comment: Please provide complete DLS curve of nanobots, not simple data.

7. *The movement of the Nanobots in different media (especially in blood) should be provided in the form of video (not less than 20 nanobot in each media). Clearly marked trajectories of the each nanobot should be provided. And the detailed motion analysis of the nanobots such as MSD should be provided as well.*

Response: As suggested by the reviewer, we have now included real time tracking trajectories of multiple MFNs in PBS, DMEM and Blood serum in the supporting information section (Figure S4).

Supplementary Figure S4. Analysis of the motion behavior of MFN. Representative tracking trajectories of MFNs in different biologically relevant media.

Comment: Only five nanobots' trajectory in every medium were provided, and it is still lack of motion analysis of the nanobots like MSD. Blood and Serum are two different mediums, the nanobots were used to capture the CTCs in blood (Fig. 4), so it would be better to provide the nanobots' trajectory in blood environment.

8. The pictures in Figure 4 and 6 are of poor quality, and only one or two cancer cells are provided. And the background interference is too difficult to judge whether cancer cells have been successfully captured accurately. Pictures with higher quality and more cancer cells should be provided.

Response: As suggested by the reviewer, we have now added images of high quality with less background interference. We have mentioned in the manuscript that only 1 to 5 CTCs per 1 mL blood sample of cancer patients were found. Hence, we could only show very few cancer cells in Figure 4 and 6.

9. *The language and grammar should be further polished.*

Response: As suggested by the reviewer we have done the necessary corrections.

Water-Powered Self-Propelled Magnetic Nanobot for Rapid and Highly Efficient Capture of Circulating Tumor Cells

Submission ID: COMMSCHEM-21-0190

Response to referees' comments

Reviewer #1

Thank you very much for the detailed response to the reviewers. However, one of my previous concerns that "nanobots are overengineered for ex vivo clinical tests as many simple nanoparticles can perform as well for ex vivo tests" has not been addressed. I think the authors can consider doing a simple experiment to compare the effectiveness of nanoparticles with no propulsion and the nanobots in CTC capture to check if there is significant benefit in using nanobots.

Response: The authors thank the reviewer for the critical comment which we feel helps us better emphasize our evidenced hypothesis and inferences. Here, we have done a detailed study and have compared the results of CTC capture by propelling nanobot and non-propelling nanobot. The results clearly show that the CTC capture efficiency of propelling nanobot is significantly higher than the non-propelling nanobot. A data table has been provided in the supporting information (Supplementary Table 2), for clarity. We wish to note here that the concept of self-propelled or ambulatory nanoparticles has risen from multiple past studies by this group, notably one we published in Banerjee et al, *Nanoscale* 2015. The study clearly demonstrated that the cancer cell capture was only ~ 22% for the simple non-propelling transferrin conjugated CNT (Tf-CNT) particle. In comparison, the cancer cell (HCT116 cells) capture efficiency of self-propelling Tf-CNT-Fe₃O₄ particle was significantly higher. This has led us to develop the present hypothesis, which reiterates the need for a particle in active motion within the bound volume for maximum collision potential and hence a heightened probability of molecular interactions leading to selective capture of biomarker-bearing cells. Our data show the unequivocal superiority of self-propelled particles in this context over to other, static models.

Supplementary Table 2. Number of HCT116 and MCF7 cells captured with MFNs in various biological media.

Cells Added	NanoSystem Used								Media
	MFN (Tf)		MFN (Ab)		MFN (Tf) + NaHCO ₃		MFN (Ab) + NaHCO ₃		
	% Cells Captured								
	HCT116	MCF7	HCT116	MCF7	HCT116	MCF7	HCT116	MCF7	DMEM
10	25	29.17	40	45.83	100	100	100	100	
25	29.17	50.00	31.25	55.36	100	100	100	100	

50	48.04	36.44	46.08	39.83	99.04	100	99.04	100	
75	46.58	42.11	43.84	48.68	97.95	100	97.95	100	
100	37.86	47.14	35.92	50.00	93.81	100	93.30	100	
10	36.11	39.29	37.5	32.14	100	100	100	100	PBS
25	38.37	57.14	36.67	48.22	100	100	100	100	
50	55.67	55.66	52.54	47.17	95.69	100	100	100	
75	31.08	51.32	35.14	55.92	96.53	100	95.14	100	
100	25.21	51.38	29.21	40.83	92.07	100	91.01	100	
10	50	68.75	44.44	50	100	100	100	100	Blood
25	66.67	56.90	64.81	41.38	100	100	100	100	
50	43.86	50.00	39.47	38.46	98.18	100	99.09	100	
75	56.76	51.33	41.22	44.00	99.31	100	98.61	100	
100	50.49	54.95	56.80	48.52	98.44	100	98.96	100	
10	40	59.09	35	50	100	100	100	100	Lysed Blood
25	58.93	52.00	41.07	38	100	100	100	100	
50	54.81	54.81	66.35	35.58	98.98	100	98.98	100	
75	41.22	52.00	45.95	45.33	98.70	100	98.70	100	
100	45.75	63.27	44.81	50	97.89	100	98.86	100	

Banerjee et al., Self-propelled carbon nanotube based microrockets for rapid capture and isolation of circulating tumor cells, *Nanoscale*, 2015 7(19), 8684-8.

Reviewer #2

1. *It would be better to compare the nanobots' speed, capture efficiency and concentration of cell in this paper with other researches in the form of Table, and it not sufficient to compare the capture time simply. In addition, it is meaningless to compare capture efficiency without providing the concentration of cell. At the same time, the paper lacks the detailed characterization of nanobots' speed.*

Response: We thank the reviewer for the critical comment. We have now given the capture efficiency and other details of recently reported nanoparticles for CTC capture as suggested by the reviewer. The table now contains cell lines used, cell count utilized during the experiments, incubation time and capture efficacy.

Supplementary Table 3. Number of cancer cells captured by recent typical nanomaterial-based CTC isolation approaches.

Sr. No	Nanoparticle	Cell line used for testing	Cell count utilized for capture efficiency	Incubation Time	Capture efficiency	Reference
1	Anti-EpCAM-modified F-MNPs and dual (EpCAM-modified and anti-N-cadherin-modified) antibody-modified F-MNPs	MCF-7	10-200 cells/mL	>20 min	97% with Anti-EpCAM-modified F-MNPs and 98.8% with dual antibody-modified F-MNPs	[1]
2	Anti-EpCAM antibody modified RBC-IMNs (Immunomagnetic micro/nanoparticles)	MCF-7 and PC-3	10-200 cells/mL	0-24 h	~90% in PBS, ~60% in blood	[2]
3	MNPs@hydrogel-anti-EpCAM nanoparticles	MCF-7	5-100 cells/mL	25 min	97% in PBS and 96% in blood	[3]
4	Gelatin nanoparticle-coated silica microbead functionalized with anti-EpCAM and anti-CD146 Abs	MCF-7, MDA-MB-231, HCT116 and HT-29	20-250 cells/mL	20 min	>80%	[4]
5	Hollow glass microspheres modified anti-EpCAM antibody	MCF7	30-1000 cells/ml in 5x diluted blood	20 min	93.6%	[5]

2. *Although the data of Zeta and FTIR was used to demonstrate each step of the modification, it still lack of the TEM figure contains multiple particles.*

Response: We thank the reviewer for the suggestion. We have now given in the Supporting Information (**Supplementary Fig. 1**) the TEM image showing multiple nanoparticles.

Supplementary Figure S1. TEM image of MFN.

3. *How long is the life of the nanobots in this paper? will the nanobots deplete within 5 minutes? It would be more reasonable to provide the video which record the corresponding motion of the nanobots.*

Response: We have studied the propulsion time of MFN in presence and absence of NaHCO_3 and have given the data in the Supporting Information (Supplementary Table. 1). As suggested by the reviewer, we have also added a video of the MFN propelling in DMEM (Supplementary Video 1).

Supplementary Table 1. Total propulsion time of MFN in various media.

NaHCO ₃ Conc (M).	MNF propulsion time (min)	
	0	0.5
DMEM	1.1	5.2
PBS	0	5.8
Serum	7	>28

4. *Please provide complete DLS curve of nanobots, not simple data.*

Response: As suggested by the reviewer, we have added the DLS curve in the Supporting Information (Supplementary Fig. S3).

Supplementary Figure S3. Dynamic light scattering (DLS) size distribution curve of MFN.

5. *Only five nanobots' trajectory in every medium were provided, and it is still lack of motion analysis of the nanobots like MSD. Blood and Serum are two different mediums, the nanobots were used to capture the CTCs in blood (Fig. 4), so it would be better to provide the nanobots' trajectory in blood environment.*

Response: As suggested by reviewer, we have added MSD to the speed data given in the manuscript text. The sentences with MSD now read as:

‘The nascent bubbles produce a strong momentum that propelled the nanobot upward with speeds of $0.815 \pm 0.086 \text{ mm s}^{-1}$, $0.707 \pm 0.06 \text{ mm s}^{-1}$ and $0.393 \pm 0.07 \text{ mm s}^{-1}$ in PBS, DMEM, and serum, respectively in presence of 1.0M NaHCO_3 .’

‘The speed of MFN increased from $0.457 \pm 0.073 \text{ mm s}^{-1}$ to $0.815 \pm 0.086 \text{ mm s}^{-1}$, $0.468 \pm 0.07 \text{ mm s}^{-1}$ to $0.707 \pm 0.06 \text{ mm s}^{-1}$ and $0.329 \pm 0.10 \text{ mm s}^{-1}$ to $0.393 \pm 0.07 \text{ mm s}^{-1}$ when the NaHCO_3 concentration is increased from 0.25 M to 1.0 M.’

Also, we have now given nanobots' trajectory in blood environment in the Supporting Information (Supplementary Fig. S6).

Supplementary Figure S6. Analysis of the motion behavior of MFN. Representative tracking trajectories of MFNs in different biologically relevant media.

REVIEWERS' COMMENTS:

Reviewer #2 (Remarks to the Author):

The article can be accepted now.